# Observer-Based Active Control Strategy for Networked Switched Systems against Two-Channel Asynchronous DoS Attacks

**Jiayuan Yin [1] and Anyang Lu [1,2,*]**

1    The College of Information Science and Engineering, Northeastern University, Shenyang 110819, China; 2270829@stu.neu.edu.cn

2    The State Key Laboratory of Synthetical Automation of Process Industries, Northeastern University, Shenyang 110819, China

\*    Correspondence: luanyang@ise.neu.edu.cn

**Abstract:** This paper addresses the security issue of networked switched systems under two-channel asynchronous denial-of-service (DoS) attacks, where the measurement channel and the control channel are subject to DoS attacks independently. For the case of partial-state measurements, an observer-based active control strategy is proposed to mitigate the negative impact on the control performance and stability of the system caused by the attacks. In this strategy, a novel mode-dependent finite-time observer is designed to estimate the system state rapidly and accurately, the predictor and the buffer are designed to ensure that the control signals transmitted to the actuator can be updated even when the control channel is blocked. Compared to the earlier results on the active control strategy that only consider the case of full-state measurements and assume that the DoS signals followed specific patterns, our work only limits the frequency and duration of the DoS signals, which is more general and challenging. Furthermore, the switching signal is designed to ensure the input-to-state stability (ISS) of the networked switched system with the active control strategy under two-channel asynchronous DoS attacks and asynchronous switching behaviors. Finally, the effectiveness and the merits of our work are validated through an example and a comparative experiment.

**Keywords:** networked switched systems; denial-of-service; asynchronous switching; active control strategy; finite-time observer





## 1. Introduction

With the combination and development of sensing, computing and communication technologies, cyber-physical systems (CPSs), i.e., systems whose physical processes are sensed and controlled by networks and remote computers [1,2], have found practical applications in many fields, such as smart grids [3–5], autonomous driving and intelligent industrial systems [6]. However, the security issue of CPSs has also attracted a lot of attention due to the use of remote computers and networks, where the external attacks introduced through the networked feedback channels may lead to deterioration of the control performance or even lead to instability [7].

The main types of cyber attacks that have been studied against CPSs include deception attacks and denial-of-service (DoS) attacks. Unlike deception attacks, DoS attacks are more practical and common as they disrupt the control performance and stability of the physical processes by blocking networked control channels and measurement channels without model knowledge and disclosure information [8,9]. Many resilient control theories have been proposed against DoS attacks on CPSs, such as DoS detection [10,11], secure state estimation [12–16], stability analysis [17–20], robust design [21–23] and so on.

Modelling of the DoS attacks is crucial for the analysis of resilience control. In [24], the DoS attack pattern is assumed to obey the Bernoulli distribution. In [25,26], the Pulse-Width Modulated DoS attacks have been studied. In [27], DoS signals with minimum sleeping time have been studied. More generally, in [17], an average dwell time (ADT)-like method is proposed to constrain the duration and frequency of DoS attacks without assuming that the DoS signals obey a certain pattern. In this paper, we will follow this DoS signal description and assume that the attack patterns employed by the adversaries are unknown.

In addition, DoS attacks against different networked channels have been widely studied. In [12,21,28,29], it is assumed that only the measurement channel is subject to DoS attacks. In [17,30], it is assumed that DoS attacks synchronously affect both the measurement and the control channels. In [31], CPSs with multiple transmission channels under DoS attacks are studied. However, asynchronous DoS attacks on both the measurement channel and the control channel are more challenging and have not received much attention.

On the other hand, as a special class of hybrid systems, networked switched systems, have been systematically studied in recent years [32–34]. In [32], the stability analysis is studied for the sampled-data switched systems with static quantizers by using the CLF method. In [33], the issue of stabilizing the switched and hybrid system with an encoding and control strategy is investigated. In [34], the adaptive tracking control problem of uncertain hybrid switching Markovian systems is studied by using the SMPLF method.

Additionally, the security issues of the networked switched systems under DoS attacks are studied in [35–38]. Specifically, in [35], the stability of networked switched systems with a ZOH controller under two-channel synchronous DoS attacks is investigated. In [36], the ISS problem of switched linear systems with unstabilizable modes is further investigated. However, it is shown that the passive defense strategies proposed in the above works lead to conservative results and strict constraints on DoS signals. In [37], the stability of the discrete switched system based on an active control strategy under two-channel asynchronous DoS attacks is investigated. This paper focuses on the scenario without any disturbances. However, in practical applications, disturbances are prevalent and can significantly impact the prediction accuracy in the active control strategy. This can, in turn, undermine the assurance of system robustness. In [38], the active control strategy is designed for switched systems suffering from disturbances and asynchronous attacks. However, the ISS conditions of the system are still not investigated.

Furthermore, observer design has been extensively studied in both networked switched systems [39,40] and CPSs [21,41,42]. Specifically, in [41], an observer-based ETC architecture is proposed for CPSs under DoS attacks, which necessitates sensors and actuators with specific computational capabilities. In [21], the robust design incorporating a finite-time observer and a predictor is proposed to rapidly estimate the system state during the sleeping periods of DoS attacks on the measurement channel. In [42], observer-based event-triggered control is studied for the continuous networked system subject to DoS attacks. In [39], a mode-dependent Luenberger-type observer is proposed for switched systems in both continuous-time and discrete-time contexts. In [40], an adaptive neural network observer is designed for networked switched systems via quantized output signals. However, to the best of our knowledge, the security issue for observer-based resilient control of networked switched systems under two-channel asynchronous DoS attacks has not been fully investigated, which motivates our work.

The main contributions of this paper are as follows:

- An observer-based active control strategy is proposed for networked switched systems, which exhibits resilience and robustness against two-channel asynchronous DoS attacks and asynchronous switching behaviors. In addition, the buffer size design approach is proposed.
- The switching signals are designed to ensure the ISS of the networked switched systems under active control strategy against two-channel asynchronous DoS attacks and asynchronous switching behaviors, specifically, the quantitative relationship between the frequency and duration of two-channel DoS attacks and the switching

frequency is revealed under ISS conditions. The results can be degraded to the non-switched system case.

- Unlike [26,37,38,43] that only consider the case of full-state measurements, the case of partial-state measurements is studied, and a mode-dependent finite-time observer is designed to rapidly and accurately estimate the system state. In addition, external disturbances are also considered in our work.
- In the existing methods [37,38,43], the effectiveness of active control strategies often relies on assuming that the DoS signals adhere to specific patterns. However, in our work, the effectiveness of the active control strategy is only related to the frequency and duration of DoS attacks, without making any assumptions about specific patterns for DoS attacks, which is more general and challenging.

The remainder of the paper is organized as follows. In Section 2, the problem statement and preliminaries are presented. The observer-based active control strategy is proposed in Section 3. The ISS analysis of the networked switched systems under two-channel asynchronous DoS attacks and asynchronous switching behaviors is given in Section 4. Section 5 gives an example and a comparative experiment to validate the availability and merits of our work. Section 6 concludes the work of this paper.

Notations: $\mathbb{N}$ and $\mathbb{R}$ denote the integer set and real number set, respectively. Given an integer (real number) $\alpha$, denote $\mathbb{N}_{\geq \alpha}$ ($\mathbb{R}_{\geq \alpha}$) be the set of integers (reals) which are not smaller than $\alpha$. Given a vector $v \in \mathbb{R}^n$, $\| v \|$ denotes its Euclidean norm. Given a matrix M, $\| M \|$ denotes its spectral norm. $U \setminus V$ denotes the set of all elements belonging to set U, but not to set V. $I$ stands for the identity matrix. $\mu_A$ denotes the logarithmic norm of mitrix A. $\overline{\lambda}(Q)$ and $\underline{\lambda}(Q)$ represent the maximum and minimum eigenvalues of matrix Q, respectively.

## 2. Problem Statement and Preliminaries

### 2.1. Networked Switched Linear System

The system to be investigated is a networked switched linear system described by

$$\begin{cases} \dot{x}(t) = A_{\sigma(t)}x(t) + B_{\sigma(t)}u(t) + w(t) \\ y(t) = C_{\sigma(t)}x(t) \end{cases} \tag{1}$$

where $x(t) \in \mathbb{R}^n$, $t \in \mathbb{R}_{\geq t_0}$ denotes the state vector, $u(t) \in \mathbb{R}^m$ denotes the control input, $w(t) \in \mathbb{R}^w$ stands for the unknown bounded external disturbance. $\sigma(t) : [t_0, \infty) \to \mathcal{M} = \{1, 2, \ldots, m\}$ denotes the switching signal which is a right-continuous piecewise constant function and $A_i$, $B_i$, $C_i$ with $i \in \mathcal{M}$ denote constant matrices with suitable dimensions. In this paper, we assume that $(A_i, B_i)$ is stabilizable, $(C_i, e^{A_i \Delta})$ is $\mu_i$-steps observable and that a state-feedback matrix $K_i$ has been predesigned such that $\Phi_i = (A_i + B_i K_i)$ is hurwitz. At all switching instants, there is no state jump.

In this paper, we assume that both the measurement channel and the control channel are networked and adopt the same sampling scheme synchronously with a fixed sampling period $\Delta$. Let $\{t_k\}_{k \in \mathbb{N}_{\geq 0}}$ denote the sequence of sampling instants, i.e.,

$$t_{k+1} - t_k = \Delta, \quad k \in \mathbb{N}_{\geq 0}. \tag{2}$$

The system's mode can switch at any $t_s(n) \geq t_0$, where $n \in \mathbb{N}_{\geq 1}$ represents the nth switching, then the sensor transmits the system's mode to the observer at the first successful transmission after $t_s(n)$. Next we introduce some assumptions used in this paper.

**Assumption 1.** *[44] (switching frequency). Let $n_\sigma(t_0, t)$, $t > t_0$ be the number of switching behaviors over the interval $[t_0, t)$. If*

$$n_\sigma(t_0, t) \leq N_\sigma + \frac{t - t_0}{\tau_\sigma} \tag{3}$$

with $N_\sigma \geq 1$, then $\tau_\sigma > \Delta$ is called the ADT. for $\tau_d \in \mathbb{R}_{>0}$, If any two consecutive switching instants $t_s(n)$ and $t_s(n+1)$ satisfy $t_s(n+1) - t_s(n) \geq \tau_d$, then $\tau_d < \tau_\sigma$ is called the minimum dwell time.

**Remark 1.** *If $\tau_a \leq \Delta$, then the switches occurring right after each sampling instant is allowed, the closed-loop system will always be mismatched and the system will become unstable.*

*2.2. Two-Channel Asynchronous DoS Attacks*

In this paper, we assume that both the measurement channel and the control channel are subject to DoS attacks. Both the mode signal and the measurement/control signal can be blocked by DoS attacks. Due to the unknown attack strategy applied by the attacker, we only limit the frequency and duration of the two-channel attacks.

We define $h_n^o$ as the sequence of DoS off/on transitions, and $\Gamma_n^o$, $n \geq 0$ as the nth DoS interval on channel $o \in \{m, c\}$. where "m" denotes the measurement channel and "c" denotes the control channel. Given $t_0, t \in \mathbb{R}_{\geq t_0}$, let

$$\Xi^o(t_0, t) := \bigcup_{n \in \mathbb{N}_{\geq 0}} \Gamma_n^o \bigcap [t_0, t] \tag{4}$$

and let

$$\Theta^o(t_0, t) := [t_0, t] \setminus \Xi^o(t_0, t) \tag{5}$$

denote the union of time intervals over $[t_0, t]$ during which the o channel is subject to DoS attacks and not subject to DoS attacks , respectively.

**Assumption 2.** *[17] (DoS duration). There exist $\kappa_{\mathcal{D}}^o \in \mathbb{R}_{\geq 0}$ and $T_{\mathcal{D}}^o \in \mathbb{R}_{>1}$ such that*

$$| \Xi^o(t_0, t) | \leq \kappa_{\mathcal{D}}^o + \frac{t - t_0}{T_{\mathcal{D}}^o} \tag{6}$$

*for all $t_0, t \in \mathbb{R}_{\geq t_0}$.*

**Remark 2.** *$T_\sigma > 1$ ensures that the DoS duration cannot be infinitely long.*

**Assumption 3.** *[17] (DoS frequency). Let $n_{\mathcal{D}}^o(t_0, t)$, $t > t_0$ be the number of DoS attacks over the interval $[t_0, t)$. If*

$$n_{\mathcal{D}}^o(t_0, t) \leq N_{\mathcal{D}}^o + \frac{t - t_0}{\tau_{\mathcal{D}}^o} \tag{7}$$

*with $N_{\mathcal{D}}^o \geq 1$ and $\tau_{\mathcal{D}}^o > \Delta$.*

*2.3. Control Objective*

**Definition 1.** *[45] System under unknown bounded external disturbance $w(t)$ is defined as input-to-state stable (ISS) if there exist some $\chi \in \mathcal{KL}$-function and $v \in \mathcal{K}_\infty$-function such that*

$$\| x(t) \| \leq \chi(\| x(t_0) \|, t) + v(\| w_t \|_\infty) \tag{8}$$

*holds for all $t \geq t_0$. $\| w_t \|_\infty := ess \, sup_{s \in [t_0, t)} \| w(s) \|$.*

The objective of our work is to design an observer-based active control strategy that exhibits resilience against DoS attacks on both the measurement channel and the control channel, thereby ensuring the ISS of the switched system.

## 3. Observer-Based Active Control Strategy

### 3.1. Mode-Dependent Finite-Time Observer Design

Let $\{w_j\}_{j\in\mathbb{N}_{\geq 0}}$ stand for the sequence of successful transmission attempts on measurement channel and let $\{z_m\}_{m\in\mathbb{N}_{\geq 0}}$ stand for the sequence of successful transmission attempts preceded by $\mu - 1$ consecutive $w_j$ with the same mode i. the observer scheme is designed as follows:

$$\begin{cases} \dot{\xi}(t) = A_{\sigma(w_{j(t)})}\xi(t) + B_{\sigma(w_{j(t)})}u(t), & t \neq w_j \\ \xi(t) = \xi(t^-) + L_{\sigma(w_{j(t)})}\left(y(t) - C_{\sigma(w_{j(t)})}\xi(t^-)\right), & t = w_j \end{cases} \tag{9}$$

where

$$\xi(t_0) = \begin{cases} L_{\sigma(w_0)}y(t_0), & t_0 = w_0 \\ \xi(z_{-1}) = 0, & otherwise \end{cases} \tag{10}$$

and

$$j(t) = \begin{cases} -1, & if\ \Theta(t_0,t) = \varnothing \\ sup\{j \in \mathbb{N}_{\geq 0} \mid w_j \in \Theta(t_0,t)\}, & otherwise \end{cases}$$

$\xi(t)$ in (9) represents the state of the finite-time observer, the initial condition is given by $\xi(z_{-1}) = 0$. Without loss of generality, we assume that at $t_0$, the observer's mode may not necessarily match the system's mode, i.e., $\sigma(w_{-1}) = \sigma(z_{-1})$ can be any $i \in \mathcal{M}$. There exists a state reconstruction matrix $L_i$ such that $R_i^{\mu_i} = 0$ holds, where $\mu_i$ denotes the observability index of $(C_i, e^{A_i\Delta})$ and $R_i = (I - L_iC_i)e^{A_i\Delta}$.

### 3.2. Active Control Strategy

In the case of DoS attacks on the control channel, the existing works usually employ a ZOH controller [17,35,36], where the actuator maintains the previous control signal when the control channel is blocked by DoS. This paper proposes an observer-based active control strategy for switched systems under two-channel asynchronous DoS attacks, ensuring that the actuator can update the control signal at each sampling instant (cf. Figure 1).

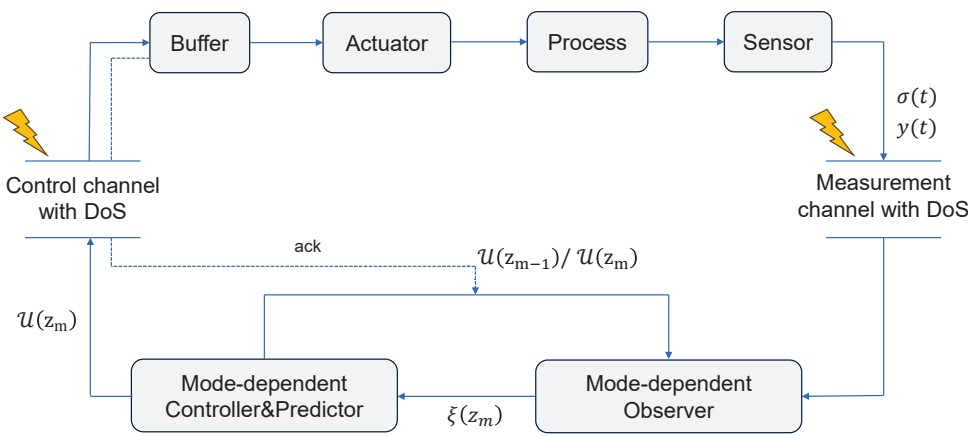

**Figure 1.** Observer-based active control strategy.

Denote $\{s_r\}_{r\in\mathbb{N}_{\geq 0}}$ as the sequence of successful transmission attempts on the control channel, and let $\{s_m\}_{m\in\mathbb{N}_{\geq 0}}$ denotes the sequence of the first $s_r$ following $z_m$ (cf. Figure 2).

The algorithm of the active control strategy can be described as follows:

- Step 1: The observer resets the estimate of the system state $\xi(z_m)$ when it receives $\mu$ consecutive measurement signals with the same mode i from the sensor, and then sends it to the predictor.
- Step 2: Based on the estimate $\xi(z_m)$, the system's mode $\sigma(z_m)$ and the actual control signals, the predictor predicts the system state $\hat{x}$ and the controller generate the control sequence $\mathcal{U}(z_m)$ and transmit it to the buffer at $s_m$ by using one data packet.

- Step 3: The buffer discards the outdated control signals and sequentially sends the control signals to the actuator, one by one, at each sampling instant. The actuator holds the control signal until the next sampling instant. Return to Step 1.

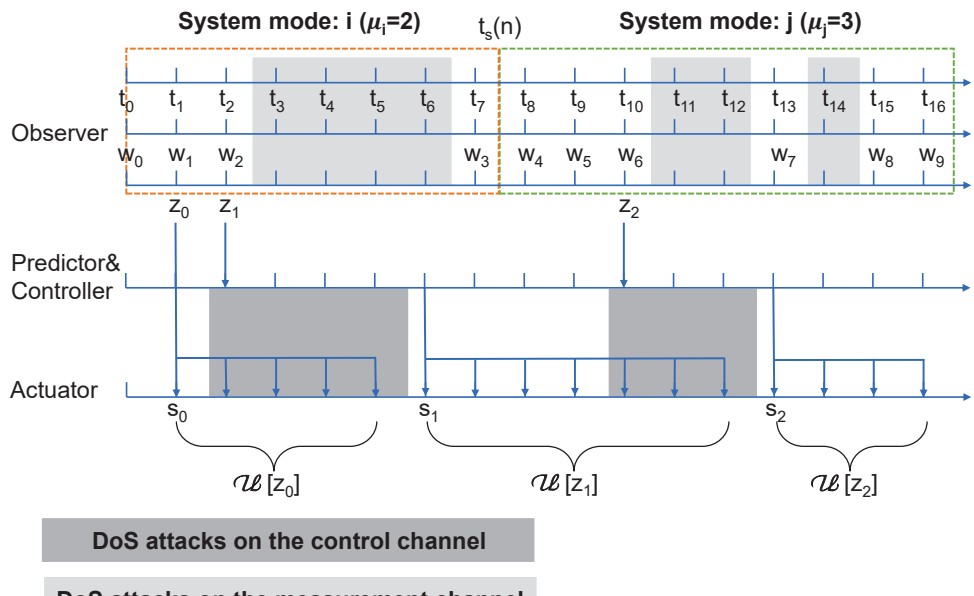

**Figure 2.** The schematic diagram of the transmission policy under switching behaviors and two-channel asynchronous DoS attacks. The solid arrows represent the successful transmissions.

The control sequence $\mathcal{U}(z_m)$ can be expressed by

$$\mathcal{U}(z_m) = \{u[z_m \mid z_m], u[z_m + \Delta \mid z_m], \ldots, u[z_m + (\mathcal{H} - 1)\Delta \mid z_m]\} \tag{11}$$

with $u[z_m + n\Delta \mid z_m] = K_{\sigma(z_m)}\hat{x}(z_m + n\Delta \mid z_m)$, $n \in [0, \mathcal{H} - 1]$ for all $m \in \mathbb{N}_{\geq 0}$. $\mathcal{H}$ represents the buffer size that will be designed later. $\hat{x}(z_m + n\Delta \mid z_m)$ represents the prediction of $x(z_m + n\Delta)$ based on $\xi(z_m)$ and $\sigma(z_m)$, and can be calculated iteratively through

$$\begin{cases} \hat{x}(t_k + \Delta \mid z_m) = A^\delta_{\sigma(z_m)}\hat{x}(t_k \mid z_m) + B^\delta_{\sigma(z_m)}u(t_k), t_k \in [z_m, s_{m+1}) \\ \hat{x}(z_m \mid z_m) = \xi(z_m) \end{cases} \tag{12}$$

where the actual control signal $u(t_k)$ can be described as

$$u(t_k) = \begin{cases} u[t_k \mid z_{m-1}] = K_{\sigma(z_{m-1})}\hat{x}(t_k \mid z_{m-1}), & t_k \in [z_m, s_m) \\ u[t_k \mid z_m] = K_{\sigma(z_m)}\hat{x}(t_k \mid z_m), & t_k \in [s_m, z_{m+1}) \end{cases}$$

for all $m \in \{-1\} \bigcup \mathbb{N}_{\geq 0}$, where $A^\delta_i = e^{A_i\Delta}$ and $B^\delta_i = B_i \int_0^\Delta e^{A_i s} ds$. The initial condition is given by $\hat{x}(t_0 \mid z_{-1}) = \hat{x}(z_{-1} \mid z_{-1}) = \xi(z_{-1}) = 0$.

After the buffer receives the new control sequence $\mathcal{U}(z_m)$, it sends an acknowledgment (ack) signal back to the controller through the control channel. Upon receiving the ack signal, the controller sends the new control sequence to the observer and the predictor to ensure that the observer, the predictor and the system adopt the same control signal at all sampling instants.

**Remark 3.** *In the time interval $[z_m, s_m)$, the system applies the preceding control sequence $\mathcal{U}(z_{m-1})$. Due to the uncertainty of the control channel's availability, the prediction of the system can only be generated one by one at each sampling instant. At $s_m$, the remaining control signals of the control sequence $\mathcal{U}(z_m)$ are generated all at once.*

### 3.3. Buffer Size Design

In the measurement channel, both DoS attacks and the switching behaviors can disrupt the estimation of the system state by the observer. Furthermore, in the control channel, DoS attacks prevent the transmission of control sequences based on the latest estimate to the buffer. Therefore, establishing the sufficient conditions for the control mechanism to generate control signals and transmit them to the buffer within a finite time is crucial for buffer size design and system ISS analysis.

First, the sufficient conditions are proposed for the mode-dependent finite-time observer to estimate the system state under DoS attacks on the measurement channel and switching behaviors.

**Lemma 1.** *Consider the networked switched linear system* (1) *under the transmission policy* (2) *and the active control strategy* (9) *and* (11), *consider the DoS signal satisfying* (6) *and* (7), *the swicthing signal satisfying* (3), *the observer can necessarily estimate the system state within a finite time if*

$$1 - \frac{1}{T_{\mathcal{D}}^m} - \frac{\mu\Delta}{\tau_{\mathcal{D}}^m} - \frac{\mu\Delta}{\tau_\sigma} > 0 \tag{13}$$

*holds true, where* $\mu = max\{\mu_i\}, i \in \mathcal{M}$. *And* $\{z_m\}_{m\in\mathbb{N}_{\geq 0}}$ *satisfies*

$$\begin{cases} z_{m+1} - z_m \leq \mathcal{Q}^m + \Delta \\ z_0 \leq t_0 + \mathcal{Q}^m + (\mu - 1)\Delta \end{cases} \tag{14}$$

*where* $\mathcal{Q}^m = (\kappa_{\mathcal{D}}^m + N_{\mathcal{D}}^m \mu\Delta + N_\sigma \mu\Delta)(1 - \frac{1}{T_{\mathcal{D}}^m} - \frac{\mu\Delta}{\tau_{\mathcal{D}}^m} - \frac{\mu\Delta}{\tau_\sigma})^{-1}$.

**Proof.** Define $\tilde{\Theta}^m(h_n^m, t)$ as the time interval starting from $h_n^m$ during which the observer is capable of generating $\xi(z_m)$ at each sampling instant. Prolonging each DoS interval and each asynchronous switching interval by $\mu$ samplings, we have

$$\begin{aligned} | \tilde{\Theta}^m(h_n^m, t) | &\geq t - h_n^m - | \Xi^m(h_n^m, t) | - n_{\mathcal{D}}^m(h_n^m, t)\mu\Delta - n_\sigma(h_n^m, t)\mu\Delta \\ &\geq t - h_n^m - (\kappa_{\mathcal{D}}^m + \frac{t - h_n^m}{T_{\mathcal{D}}^m}) - (N_{\mathcal{D}}^m + \frac{t - h_n^m}{\tau_{\mathcal{D}}^m})\mu\Delta - (N_\sigma + \frac{t - h_n^m}{\tau_\sigma})\mu\Delta \\ &\geq (t - h_n^m)(1 - \frac{1}{T_{\mathcal{D}}^m} - \frac{\mu\Delta}{\tau_{\mathcal{D}}^m} - \frac{\mu\Delta}{\tau_\sigma}) - \kappa_{\mathcal{D}}^m - N_{\mathcal{D}}^m \mu\Delta - N_\sigma \mu\Delta \end{aligned} \tag{15}$$

$| \tilde{\Theta}^m(h_n^m, t) | > 0$ implies that within the time interval $[h_n, t]$, there must be at least $\mu$ consecutive successful transmissions with the same mode passing through the measurement channel and being received by the observer. From (15) we can see that, if (13) holds true, then there must exist a sufficiently large t such that $| \tilde{\Theta}^m(h_n^m, t) | > 0$ is satisfied.

Then, if switching signals or DoS attacks do not occur in $[t_0, t_0 + (\mu - 1)\Delta]$, then $z_0 \leq t_0 + \mathcal{Q}^m + (\mu - 1)\Delta$ holds trivially. If switching signals or DoS attacks occur in $[t_0, t_0 + (\mu - 1)\Delta]$, we have $\mu$ consecutive successful transmissions with the same mode no later than $t_0 + \mathcal{Q}^m + (\mu - 1)\Delta$. Similarly, if switching signals or DoS attacks do not occur in $[z_m, z_m + \Delta]$, then $z_{m+1} - z_m \leq \mathcal{Q}^m + \Delta$ holds trivially. If switching signals or DoS attacks occur in $[z_m, z_m + \Delta]$, we have $\mu$ consecutive successful transmissions with the same mode no later than $z_m + \mathcal{Q}^m + \Delta$. □

**Remark 4.** *In* [21], *sufficient conditions are proposed for the finite-time observer to estimate the system state under DoS attacks on the measurement channel. In this paper, both DoS attacks and switching behaviors can interrupt the estimation of system state by the finite-time observer. Lemma 1 provides the limitation for DoS signals on the measurement channel and switching signals in the worst-case scenario, under which the estimation of the system state can be achieved.*

Next, the sufficient conditions are proposed for the data packet of the control sequence $\mathcal{U}(z_m)$ to be transmitted to the buffer under DoS attacks on the control channel.

**Lemma 2.** *Consider the transmission policy (2) and the DoS signal satisfying (6) and (7), the control sequence $\mathcal{U}(z_m)$ can be transmitted to the buffer within a finite time if*

$$1 - \frac{1}{T_{\mathcal{D}}^c} - \frac{\Delta}{\tau_{\mathcal{D}}^c} > 0 \tag{16}$$

*holds true. Recall that we denote $\{s_r\}_{r \in \mathbb{N}_{\geq 0}}$ as the sequence of successful transmission attempts on the control channel and $\{s_r\}_{r \in \mathbb{N}_{\geq 0}}$ satisfies*

$$\begin{cases} s_{r+1} - s_r \leq \mathcal{Q}^c + \Delta \\ s_0 \leq t_0 + \mathcal{Q}^c \end{cases} \tag{17}$$

*where $\mathcal{Q}^c = (\kappa_{\mathcal{D}}^c + N_{\mathcal{D}}^c \Delta)(1 - \frac{1}{T_{\mathcal{D}}^c} - \frac{\Delta}{\tau_{\mathcal{D}}^c})^{-1}$.*

**Proof.** The proof of Lemma 2 is similar to the proof of Lemma 1 under the conditions of $\mu = 1$ and swicthing-free. Therefore, it is omitted. □

Finally, based on Lemma 1 and Lemma 2, the minimum buffer size in the worst-case scenario is designed.

**Lemma 3.** *Consider system (1) under the transmission policy (2) and the active control strategy (9) and (11), consider the DoS signals on the measurement channel and the switching signals satisfying (13) and the DoS signals on the control channel satisfying (16). The buffer size is designed to update the control signal to the actuator at any sampling time under switching signals and two-channel asynchronous DoS attacks if*

$$\mathcal{H} \geq \frac{\mathcal{Q}^m + \mathcal{Q}^c}{\Delta} + 1 \tag{18}$$

*holds true.*

**Proof.** Recall that we denote $\{s_m\}_{m \in \mathbb{N}_{\geq 0}}$ as the sequence of the first $s_r$ following $z_m$. At $z_m$, the controller needs to calculate the control sequence from $u[z_m \mid z_m]$ to $u[s_{m+1} - \Delta \mid z_m]$ based on the value of $\xi(z_m)$. In the worst-case scenario, we assume that there are DoS attacks on the control channel right at $z_{m+1}$, then combining (14) and (17), we have

$$s_{m+1} - z_m \leq \mathcal{Q}^m + \mathcal{Q}^c + \Delta. \tag{19}$$

Therefore, as long as the buffer size meets (18), it is guaranteed that the actuator can update the control sequence $\mathcal{U}(z_m)$ at each sampling time before the next control sequence $\mathcal{U}(z_{m+1})$ based on $\xi(z_{m+1})$ is successfully transmitted to the buffer through the control channel. □

**Remark 5.** *Since the initial condition is given by $\xi(z_{-1}) = 0$, the control sequence $u[t_k \mid z_{-1}] = K_{\sigma(z_{-1})}\hat{x}(t_k \mid z_{-1}) = 0$ for all $t_k \in [t_0, s_0)$, the buffer outputs remain at 0 before $s_0$. Therefore, when designing the buffer size, there is no need to consider the time interval $[t_0, s_0]$.*

## 4. Input-to-State Stability Analysis

### 4.1. Dynamics under Two-Channel DoS Attacks without Asynchronous Switching

First, we consider the scenario without asynchronous switching. Denote $\hat{x}(t)$ as the prediction of $x(t)$ applied be the actuators. We define $\theta(t) = \hat{x}(t) - x(t)$ as the prediction error, and let $e(t) = x(t_k) - x(t)$ be the error between the state at the last sampling instant

and the state at t. Then consider any interval $[t_k, t_{k+1}]$, the dynamics of the switched system can be rewritten as

$$
\begin{aligned}
\dot{x}(t) &= \Phi_i x(t) + B_i K_i (\hat{x}(t_k) - x(t)) + w(t) \\
&= \Phi_i x(t) + B_i K_i (\hat{x}(t_k) - x(t_k) + x(t_k) - x(t)) + w(t) \\
&= \Phi_i x(t) + B_i K_i \theta(t_k) + B_i K_i e(t) + w(t).
\end{aligned}
\tag{20}
$$

First, we find the upper bound of $\theta(t_k)$. We denote $\phi(t) = \xi(t) - x(t)$ as the observation error. Considering the finite-time observer (9), we obtain

$$
\phi(z_m) = (I - L_i C_i)\phi(z_m^-),
\tag{21}
$$

combining with $\dot{\phi}(t) = A_i \phi(t) - w(t)$ and $R_i^{\mu_i} = 0$, we obtain

$$
\begin{aligned}
\phi(z_m) &= (I - L_i C_i)\left(e^{A_i \Delta}\phi(z_m - \Delta) - \int_{z_m - \Delta}^{z_m} e^{A_i(z_m - \tau)}w(\tau)d\tau\right) \\
&= R_i(I - L_i C_i)\phi((z_m - \Delta)^-) - (I - L_i C_i)\int_{z_m - \Delta}^{z_m} e^{A_i(z_m - \tau)}w(\tau)d\tau \\
&= R_i^{\mu_i - 1}(I - L_i C_i)\phi((z_m - (\mu_i - 1)\Delta)^-) \\
&\quad - (I - L_i C_i)\sum_{p=0}^{\mu_i - 2} R_i^p \int_{z_m - p\Delta - \Delta}^{z_m - p\Delta} e^{A_i(z_m - p\Delta - \tau)}w(\tau)d\tau \\
&= R_i^{\mu_i} e^{-A_i \Delta}\phi((z_m - (\mu_i - 1)\Delta)^-) \\
&\quad - (I - L_i C_i)\sum_{p=0}^{\mu_i - 2} R_i^p \int_{z_m - p\Delta - \Delta}^{z_m - p\Delta} e^{A_i(z_m - p\Delta - \tau)}w(\tau)d\tau \\
&= -(I - L_i C_i)\sum_{p=0}^{\mu_i - 2} R_i^p \int_{z_m - p\Delta - \Delta}^{z_m - p\Delta} e^{A_i(z_m - p\Delta - \tau)}w(\tau)d\tau.
\end{aligned}
\tag{22}
$$

Using the property that $\| e^{At} \| \le e^{\mu_A t}$ for all $t \in \mathbb{R}_{\ge t_0}$, we have

$$
\| \phi(z_m) \| \le \| (I - L_i C_i) \| \sum_{p=0}^{\mu_i - 2} \| R_i \|^p \frac{1}{\mu_{A_i}}(e^{\mu_{A_i}\Delta} - 1) \| w_t \|_\infty
\tag{23}
$$

$$
\le \rho_1 \| w_t \|_\infty
\tag{24}
$$

where $\rho_1 = \| (I - L_i C_i) \| \sum_{p=0}^{\mu_i - 2} \| R_i \|^p \frac{1}{\mu_{A_i}}(e^{\mu_{A_i}\Delta} - 1)$, thus $\phi(z_m)$ is upper bounded. Recall now that $\xi(z_m) = \hat{x}(z_m \mid z_m)$, we then have

$$
\| \theta(z_m \mid z_m) \| = \| \hat{x}(z_m \mid z_m) - x(z_m) \| \le \rho_1 \| w_t \|_\infty.
\tag{25}
$$

During the time interval $[z_m, s_{m+1}]$, we further have

$$
\begin{aligned}
\| \theta(t_k \mid z_m) \| &\le e^{\mu_{A_i}(t_k - z_m)} \| \theta(z_m \mid z_m) \| + \int_{z_m}^{t_k} e^{\mu_{A_i}(t_k - \tau)}w(\tau)d\tau \\
&\le \left(\rho_1 e^{\mu_{A_i}\mathcal{H}\Delta} + \frac{1}{\mu_{A_i}}(e^{\mu_{A_i}\mathcal{H}\Delta} - 1)\right) \| w_t \|_\infty \\
&\le \rho \| w_t \|_\infty
\end{aligned}
\tag{26}
$$

where $\rho = \max\limits_{i \in \mathcal{M}} \left\{ \rho_1 e^{\mu_{A_i} \mathcal{H}\Delta} + \frac{1}{\mu_{A_i}} (e^{\mu_{A_i} \mathcal{H}\Delta} - 1) \right\}$. Note that the prediction error resets at each $s_m$. We finally have

$$\| \theta(t_k) \| \leq \rho \| w_t \|_\infty . \tag{27}$$

Furthermore, in the case where no switching occurs in interval $[z_{m-1}, z_m)$, by applying the triangular inequality we have

$$\| \hat{x}(t_k \mid z_{m-1}) - \hat{x}(t_k \mid z_m) \| \leq 2\rho \| w_t \|_\infty, \quad t_k \in [z_m, s_m). \tag{28}$$

Next, we find the upper bound of $e(t)$.

**Lemma 4.** *Consider system (1) under the transmission policy (2) and the active control strategy (9) and (11), in the absence of switching, if the sampling rate is properly chosen by $\Delta = t_{k+1} - t_k \leq \overline{\Delta}$, $k \in \mathbb{N}_{\geq 0}$,*

$$\overline{\Delta} := \begin{cases} \left( \frac{\varrho}{1+\varrho} \right) \frac{1}{max\{\|\Phi_i\|, 1, \|A_i\|\}}, & \mu_{A_i} \leq 0 \\ \frac{1}{\mu_{A_i}} log \left[ \left( \frac{\varrho}{1+\varrho} \right) \frac{1}{max\{\|\Phi_i\|, 1, \|A_i\|\}} \mu_{A_i} + 1 \right], & \mu_{A_i} > 0. \end{cases} \tag{29}$$

*Then $e(t) = x(t_k) - x(t)$ satisfies*

$$\| e(t) \| \leq \varrho \| x(t) \| + \varrho(2\rho + 1) \| w_t \|_\infty, \tag{30}$$

*where $\varrho$ is designed and used to describe the disparity between the state of the system at time t and the state of the system at the most recent sampling instant before t.*

**Proof.** Consider any interval $[t_k, t_{k+1}]$, the dynamics of e(t) satisfy

$$\begin{aligned} \dot{e}(t) &= -A_i x(t) - B_i K_i \hat{x}(t_k) - w(t) \\ &= A_i \hat{x}(t_k) - A_i x(t) - A_i \hat{x}(t_k) - B_i K_i \hat{x}(t_k) - w(t) \\ &= A_i(\hat{x}(t_k) - x(t_k) + x(t_k) - x(t)) - \phi \hat{x}(t_k) - w(t) \\ &= A_i \theta(t_k) + A_i e(t) - \phi \hat{x}(t_k) - w(t). \end{aligned} \tag{31}$$

Notice that $e(t_k) = 0$, we then have

$$e(t) \leq \mathcal{N} \int_{t_k}^t e^{\mu_A(t-\tau)} (\| \theta(t_k) \| + \| \hat{x}(t_k) \| + \| w(\tau) \|) d\tau \tag{32}$$

where $\mathcal{N} = max\{\| \Phi_i \|, 1, \| A_i \|\}$, we further have

$$e(t) \leq \mathcal{N} \mathcal{F}(t - t_k)(\| \theta(t_k) \| + \| \hat{x}(t_k) \| + \| w_t \|_\infty) \tag{33}$$

where $\mathcal{F}(t - t_k) = \int_{t_k}^t e^{\mu_A(t-\tau)} d\tau$. From (27) we further have

$$\| \hat{x}(t) \| \leq \| x(t) \| + \rho \| w_t \|_\infty \tag{34}$$

then we have

$$\begin{aligned} e(t) &\leq \mathcal{N} \mathcal{F}(t - t_k)(2\rho + 1) \| w_t \|_\infty + \mathcal{N} \mathcal{F}(t - t_k)(\| x(t) \| + \| e(t) \|) \\ &\leq \frac{\mathcal{N} \mathcal{F}(t - t_k)}{1 - \mathcal{N} \mathcal{F}(t - t_k)} \| x(t) \| + \frac{\mathcal{N} \mathcal{F}(t - t_k)}{1 - \mathcal{N} \mathcal{F}(t - t_k)} (2\rho + 1) \| w_t \|_\infty \end{aligned} \tag{35}$$

By letting $\frac{\mathcal{N}\mathcal{F}(t-t_k)}{1-\mathcal{N}\mathcal{F}(t-t_k)} = \varrho$, notice that $\mathcal{F}(0) = 0$ and that $\mathcal{F}(t - t_k)$ is monotonically increasing with t, the positive correlation between $\varrho$ and $\Delta$ can be described as

$$\Delta = \frac{1}{\mu_{A_i}} log \left[ \left( \frac{\varrho}{1 + \varrho} \right) \frac{1}{max\{\parallel \Phi_i \parallel, 1, \parallel A_i \parallel\}} \mu_{A_i} + 1 \right] \tag{36}$$

for special cases where $\mu_A \leq 0$, we have

$$\mathcal{F}(\Delta) \leq \Delta. \tag{37}$$

Therefore, as long as $\Delta$ is chosen to be sufficiently small to satisfy (29), (30) is guaranteed to hold true. □

Based on (27) and Lemma 4, Given any symmetric positive definite matrix $Q_i$, let $P_i$ be the unique solution of the Lyapunov equation $\Phi_i^\mathsf{T} P_i + P_i \Phi_i = -Q_i$. Let $V_i(x) = x^\mathsf{T} P_i x$. Substituting (20) into the derivative of $V_i(x) = x^\mathsf{T} P_i x$ yields

$$\dot{V}_i(x(t)) = x(t)^\mathsf{T}(\Phi_i^\mathsf{T} P_i + P_i \Phi_i)x(t) + 2e(t)^\mathsf{T} K_i B_i^\mathsf{T} Px(t) + 2\theta(t_k)^\mathsf{T} K_i B_i^\mathsf{T} Px(t) + 2x(t)^\mathsf{T} Pw(t)$$
$$\leq (-\gamma_1 + \gamma_2 \varrho) \parallel x(t) \parallel^2 + (\gamma_2(\rho + \varrho(2\rho + 1)) + \gamma_3) \parallel x(t) \parallel v(t), \tag{38}$$

where $\gamma_1 = \underline{\lambda}(Q_i)$, $\gamma_2 = \parallel 2K_i B_i^\mathsf{T} P_i \parallel$, $\gamma_3 = \parallel 2P_i \parallel$, $v(t) = sup\{\parallel w(t) \parallel, \parallel w_t \parallel_\infty\}$. By selecting an appropriate sampling period $\Delta$, we can obtain a sufficiently small $\varrho$ to make $-\gamma_4 = (-\gamma_1 + \gamma_2 \varrho)$ negative according to Lemma 4. Using Young's inequality yields

$$\dot{V}_i(x(t)) \leq -\frac{\gamma_4}{2} \parallel x(t) \parallel^2 + \frac{(\gamma_2(\rho + \varrho(2\rho + 1)) + \gamma_3)^2}{2\gamma_4} v^2(t)$$
$$\leq -\frac{\gamma_4}{2\overline{\lambda}(P_i)} V_i(x(t)) + \frac{(\gamma_2(\rho + \varrho(2\rho + 1)) + \gamma_3)^2}{2\gamma_4} v^2(t), \tag{39}$$

finally we obtain

$$V_i(x(t)) \leq e^{-\alpha_i(t-s_m)} V_i(x(s_m)) + a_i \parallel w_t \parallel_\infty^2$$
$$\leq e^{-\alpha(t-s_m)} V_i(x(s_m)) + a \parallel w_t \parallel_\infty^2 \tag{40}$$

where $\alpha_i = \frac{\gamma_4}{2\overline{\lambda}(P_i)}$, $a_i = \frac{(\gamma_2(\rho+\varrho(2\rho+1))+\gamma_3)^2}{2\gamma_4\alpha_i}$, $\alpha = \underset{i \in \mathcal{M}}{min}\{\alpha_i\}$, $a = \underset{i \in \mathcal{M}}{max}\{a_i\}$.

### 4.2. Dynamics under Two-Channel DoS Attacks with Asynchronous Switching

Next, we consider the scenario with asynchronous switching. Based on the assumptions stated earlier, the system's mode and $\xi(z_m)$ are transmitted to the predictor at $z_m$. Therefore, if the system switches between $[z_m, z_{m+1}]$, the mode of the control sequence will mismatch with the system's mode throughout the interval $[t_s(n), s_{m+1})$, where $t_s(n) \in [z_m, z_{m+1}]$.

In the actual control process, the switching behaviors are not very frequent, so we assume that the asynchronous switching intervals do not overlap, and for simplicity, we assume that the minimum dwell time of the switching signal satisfies

$$\tau_d \geq 2(\mathcal{Q}^m + \Delta) \tag{41}$$

Due to the negative impact of asynchronous switching on system performance and stability, in the worst-case scenario, we maximize the duration of asynchronous switching within interval $[t_0, t]$; specifically, assuming that the system switches immediately after $z_m$

each time. We denote $\Pi(t_0, t)$ as the union of asynchronous intervals within $[t_0, t]$. The dynamics of the system under asynchronous switching can be described as

$$\dot{x}(t) = A_j x(t) + B_j K_i \hat{x}(t_k) + w(t). \tag{42}$$

Substituting it into the derivative of $V_j(x) = x^\mathsf{T} P_j x$ yields

$$
\begin{aligned}
\dot{V}(x(t)) &= x^\mathsf{T}(t)(A_j^\mathsf{T} P_j + P_j A_j)x(t) + 2x^\mathsf{T}(t)P_j B_j K_i \hat{x}(t_k) + 2x^\mathsf{T}(t)P_j w(t) \\
&\leq \eta_1 \parallel x(t) \parallel^2 + \eta_2 \parallel x(t) \parallel \parallel \hat{x}(t_k) \parallel + \eta_3 \parallel x(t) \parallel v(t)
\end{aligned} \tag{43}
$$

where $\eta_1 = \parallel A_j^\mathsf{T} P_j + P_j A_j \parallel$, $\eta_2 = \parallel 2P_j B_j K_i \parallel$, $\eta_3 = \parallel 2P_j \parallel$. Notice that (34) does not necessarily hold during asynchronous switching; a lemma is required to further describe the energy function of the system under asynchronous switching.

**Lemma 5.** *Consider the sampling scheme (2) satisfying (29) in Lemma 4, where $-\gamma_1 + \gamma_2 \varrho < 0$, the switching signal satisfying (41), then $\hat{x}(t_k)$ of the control sequence satisfying*

$$\parallel \hat{x}(t_k) \parallel \leq \Gamma \parallel x(z_m) \parallel + \Upsilon \parallel w_t \parallel_\infty \tag{44}$$

*where $\Gamma = \dfrac{\overline{\lambda}(P_i)}{\underline{\lambda}(P_i)}$, $\Upsilon = \dfrac{\overline{\lambda}(P_i)}{\underline{\lambda}(P_i)}\rho + \sqrt{\dfrac{\theta_2 \overline{\lambda}(P_i)}{\theta_1 \underline{\lambda}^2(P_i)}}$.*

**Proof.** Please see Appendix A. $\square$

By Lemma 5, (43) can be further rewritten as

$$\dot{V}(x(t)) \leq \eta_1 \parallel x(t) \parallel^2 + \eta_2 \Gamma \parallel x(t) \parallel \parallel x(z_m) \parallel + (\eta_2 \Upsilon + \eta_3) \parallel x(t) \parallel v(t). \tag{45}$$

It can be obtained

$$2 \parallel x(t) \parallel \parallel x(z_m) \parallel \leq \frac{\eta_1}{\eta_2 \Gamma} \parallel x(t) \parallel^2 + \frac{\eta_2 \Gamma}{\eta_1} \parallel x(z_m) \parallel^2 \tag{46}$$

and

$$2 \parallel x(t) \parallel v(t) \leq \frac{\eta_1}{\eta_2 \Upsilon + \eta_3} \parallel x(t) \parallel^2 + \frac{\eta_2 \Upsilon + \eta_3}{\eta_1} v^2(t). \tag{47}$$

Then,

$$\dot{V}(x(t)) \leq 2\eta_1 \parallel x(t) \parallel^2 + \frac{(\eta_2 \Gamma)^2}{2\eta_1} \parallel x(z_m) \parallel^2 + \frac{(\eta_2 \Upsilon + \eta_3)^2}{2\eta_1} v^2(t). \tag{48}$$

Thus,

$$\dot{V}(x(t)) \leq \beta_{ij} max\{V(x(t)), V(x(z_m)\} + \frac{(\eta_2 \Upsilon + \eta_3)^2}{2\eta_1} \parallel w_t \parallel_\infty^2 \tag{49}$$

where $\beta_{ij} = max\{\frac{2\eta_1}{\underline{\lambda}(P_j)} + \frac{(\eta_2 \Gamma)^2}{2\eta_1 \underline{\lambda}(P_j)}\}$. By iteration, we finally have

$$V(x(t)) \leq e^{\beta(t-z_m)} V(x(z_m)) + e^{\beta(t-z_m)} b \parallel w_t \parallel_\infty^2, \quad t \in \Pi(z_m, s_{m+1}) \tag{50}$$

where $\beta = \underset{i,j \in \mathcal{M}}{max}\{\beta_{ij}\}$, $b = \underset{i,j \in \mathcal{M}}{max}\{\frac{(\eta_2 \Upsilon + \eta_3)^2}{2\eta_1 \beta_{ij}}\}$.

**Remark 6.** *There are two reasons why switching after $z_m$ is the worst case; first, in this paper we consider the system under asynchronous switching as an open-loop system with an upper bounded input, and by Lemma 5 we show that the upper bound on the input is maximized when the switching occurs exactly after $z_m$. Furthermore, when the switching occurs exactly after the $z_m$, the mode of the system must wait until $z_{m+1}$ to be corrected and until $s_{m+1}$ to be applied to the actuator, thus maximising the asynchronous switching duration.*

**Remark 7.** *For the case where $z_{m+1}$ belongs to $(z_m, s_m]$, $s_m$ coincides with $s_{m+1}$, and still assuming that the system switches immediately after $z_m$. it is easy to obtain that the energy function of $x(t)$ within the asynchronous switching interval $(z_m, s_m)$ can still be described by (50). The duration of the asynchronous switching is $s_m - z_m \leq Q^c$ (this case is obviously not the worst-case-scenario).*

Now it is ready to derive the sufficient conditions for ISS of the switched systems under two-channel asynchronous DoS attacks.

*4.3. Input-to-State Stability Analysis*

**Theorem 1.** *Consider the switched linear system (1) under sampling logic (29) and observer-based active control strategy (9), (11), $\zeta V(t_s^-(n)) \geq V(t_s(n))$ with $\zeta \geq 1$. Consider the DoS signals on the measurement channel and the switching signals satisfying (13) and the DoS signals on the control channel satisfying (16). If $\tau_\sigma$ in (3) satisfy*

$$\begin{cases} \tau_\sigma > \frac{ln\zeta + (\alpha + \beta)(Q^m + Q^c + \Delta)}{\alpha} \\ \tau_\sigma > \tau_d \geq 2(Q^m + \Delta) \end{cases} \tag{51}$$

*where $\alpha$, $\beta$ are in (40) and (50), $Q^m$ and $Q^c$ are in (14) and (17), then the system is ISS.*

**Proof.** Recall that the initial condition is given by $\xi(z_{-1}) = 0$, the control sequence $\mathcal{U}(z_{-1}) = 0$ over the time interval $[t_0, s_0]$. From Lemma 5 it is easy to see that (44) holds trivially and the energy function over $[t_0, s_0]$ can be expressed as (50). Combining (40) and (50), by iteration, the energy function over $[t_0, t]$ can be derived as

$$\begin{aligned} V(x(t)) &\leq \zeta^{n_\sigma(t_0, t)} e^{-\alpha(t-t_0)} e^{(\alpha+\beta)|\Pi(t_0,t)|} V(x(t_0)) \\ &\quad + \zeta^{n_\sigma(t_0, t)} e^{-\alpha(t-t_0)} e^{(\alpha+\beta)|\Pi(t_0,t)|} b \parallel w_t \parallel_\infty^2 \\ &\quad + max\{a, b\} \left(1 + 2 \sum_{i=1}^{n_\sigma(t_0, t)} \zeta^{n_\sigma(t_s(i), t)} e^{-\alpha(t-t_s(i))} e^{(\alpha+\beta)|\Pi(t_s(i), t)|}\right) \parallel w_t \parallel_\infty^2. \end{aligned} \tag{52}$$

We first prove that the third term on the RHS of the inequality (52) is bounded. Due to Assumption 1, the summation

$$\sum_{i=1}^{n_\sigma(t_0, t)} \zeta^{n_\sigma(t_s(i), t)} e^{-\alpha(t-t_s(i))} e^{(\alpha+\beta)|\Pi(t_s(i), t)|} \tag{53}$$

can be rewritten as

$$\begin{aligned} &\sum_{i=1}^{n_\sigma(t_0, t)} e^{n_\sigma(t_s(i), t) ln\zeta - \alpha(t - t_s(i)) + (\alpha+\beta)[n_\sigma(t_s(i), t)\mathcal{H}\Delta]} \\ &\leq e^{N_\sigma ln\zeta + (\alpha+\beta) N_\sigma \mathcal{H}\Delta} \sum_{i=1}^{n_\sigma(t_0, t)} e^{[\frac{ln\zeta}{\tau_\sigma} + (\alpha+\beta)\frac{\mathcal{H}\Delta}{\tau_\sigma} - \alpha](t - t_s(i))} \\ &\leq e^{N_\sigma ln\zeta + (\alpha+\beta) N_\sigma \mathcal{H}\Delta} \sum_{i=1}^{n_\sigma(t_0, t)} e^{\epsilon(t - t_s(i))}, \end{aligned} \tag{54}$$

where $\epsilon = \frac{ln\zeta}{\tau_\sigma} + (\alpha + \beta)\frac{\mathcal{H}\Delta}{\tau_\sigma} - \alpha \leq 0$, from (3), we have

$$t - t_s(i) \geq \tau_\sigma N(t_s(i), t) - \tau_\sigma N_0, \tag{55}$$

combining (54) and (55), the summation can be further rewritten as

$$e^{N_\sigma ln\zeta + (\alpha+\beta)N_\sigma \mathcal{H}\Delta - \epsilon N_\sigma \tau_\sigma} \sum_{i=1}^{N(t_0,t)} e^{\epsilon \tau_\sigma i}$$

$$\leq e^{N_\sigma ln\zeta + (\alpha+\beta)N_\sigma \mathcal{H}\Delta - \epsilon N_\sigma \tau_\sigma} \frac{e^{\epsilon \tau_\sigma}}{1 - e^{\epsilon \tau_\sigma}} \tag{56}$$

thus the third term on the RHS of (52) is bounded.

Similarly, the first and the second terms on the RHS of the inequality (52) can be respectively rewritten as

$$e^{N_\sigma ln\zeta + (\alpha+\beta)(N_\sigma \mathcal{H}\Delta + \mathcal{Q}^m + \mathcal{Q}^c + (\mu-1)\Delta) + \epsilon(t-t_0)} V(x(t_0)) \tag{57}$$

and

$$e^{N_\sigma ln\zeta + (\alpha+\beta)(N_\sigma \mathcal{H}\Delta + \mathcal{Q}^m + \mathcal{Q}^c + (\mu-1)\Delta) + \epsilon(t-t_0)} b \parallel w_t \parallel_\infty^2 \tag{58}$$

letting $\Omega = e^{N_\sigma ln\zeta + (\alpha+\beta)N_\sigma \mathcal{H}\Delta}$, $\Omega' = e^{N_\sigma ln\zeta + (\alpha+\beta)(N_\sigma \mathcal{H}\Delta + \mathcal{Q}^m + \mathcal{Q}^c + (\mu-1)\Delta)}$ (52) can be rewritten as

$$V(x(t)) \leq \Omega' e^{\epsilon(t-t_0)} V(x(t_0))$$

$$+ max\{a, b\}(1 + \Omega' + 2\Omega e^{-\epsilon N_0 \tau_\sigma} \frac{e^{\epsilon \tau_\sigma}}{1 - e^{\epsilon \tau_\sigma}}) \parallel w_t \parallel_\infty^2. \tag{59}$$

Let $\overline{\lambda}(P_i)$ and $\underline{\lambda}(P_i)$ respectively represent the maximum and minimum eigenvalues of the positive-definite matrix $P_i, \forall i \in \mathcal{M}$. By Cauchy–Schwarz inequality, we finally obtain

$$\parallel x(t) \parallel \leq \Omega'^{\frac{1}{2}} \sqrt{\frac{\overline{\lambda}(P_i)}{\underline{\lambda}(P_i)}} e^{\frac{\epsilon}{2}(t-t_0)} \parallel x(t_0) \parallel$$

$$+ \sqrt{\frac{max\{a, b\}}{\underline{\lambda}(P_i)}}(1 + \Omega' + 2\Omega e^{-\epsilon N_\sigma \tau_\sigma} \frac{e^{\epsilon \tau_\sigma}}{1 - e^{\epsilon \tau_\sigma}})^{\frac{1}{2}} \parallel w_t \parallel_\infty. \tag{60}$$

This completes the proof. □

**Remark 8.** *In Section 5, we verify through simulation that $\beta$ tends to be much larger than $\alpha$, i.e., in (51), the average dwell time $\tau_\sigma$ is much larger than the minimum dwell time $\tau_d$. Therefore, it is not out of generality to assume that the asynchronous switching intervals do not overlap.*

In what follows, the sufficient conditions for ISS stability of non-switched systems under the active control strategy against two-channel asynchronous DoS attacks is given.

**Corollary 1.** *Consider a non-switched linear system*

$$\begin{cases} \dot{x}(t) = Ax(t) + Bu(t) + w(t) \\ y(t) = Cx(t) \end{cases} \tag{61}$$

under the active control strategy (9), (11). Consider the sampling rate chosen by sampling logic (29), where $\varrho$ is a positive constant satisfying $-\gamma_1 + \gamma_2\varrho < 0$. Then the closed-loop system is ISS for two-channel asynchronous DoS attacks satisfying

$$\begin{cases} 1 - \frac{1}{T_{\mathcal{D}}^m} - \frac{\mu\Delta}{\tau_{\mathcal{D}}^m} > 0 \\ 1 - \frac{1}{T_{\mathcal{D}}^c} - \frac{\Delta}{\tau_{\mathcal{D}}^c} > 0. \end{cases}$$

The buffer size is designed to update the control signal to the actuator at any sampling time under two-channel asynchronous DoS attacks if

$$\mathcal{H} \geq \frac{\tilde{\mathcal{Q}}^m + \mathcal{Q}^c}{\Delta} + 1 \tag{62}$$

holds true. Where $\tilde{\mathcal{Q}}^m = (\kappa_{\mathcal{D}}^m + N_{\mathcal{D}}^m\mu\Delta)(1 - \frac{1}{T_{\mathcal{D}}^m} - \frac{\mu\Delta}{\tau_{\mathcal{D}}^m})^{-1}$.

**Proof.** The proof of this corollary is similar to the derivation in Section 4.1 in the absence of switching behaviors, and is therefore omitted. □

### 5. Numerical Example

In this section, the main results of our work are verified by an example. Furthermore, a comparative experiment is presented to verify the superiority of the active control strategy over the existing ZOH controller proposed in [17,35,36] against two-channel DoS attacks.

Consider the networked switched system consisting of modes $i$ and $j$

$$A_i = \begin{bmatrix} 1 & 0 \\ 0 & -1 \end{bmatrix}, B_i = \begin{bmatrix} 1 \\ 0 \end{bmatrix}, C_i = \begin{bmatrix} 1 & 1 \end{bmatrix}, L_i = \begin{bmatrix} 5.52 \\ -4.52 \end{bmatrix}, K_i = \begin{bmatrix} -2 & 0 \end{bmatrix},$$

$$A_j = \begin{bmatrix} -1 & 1 \\ 1 & 0 \end{bmatrix}, B_j = \begin{bmatrix} 0 \\ 1 \end{bmatrix}, C_j = \begin{bmatrix} 1 & 2 \end{bmatrix}, L_j = \begin{bmatrix} 18.08 \\ -8.54 \end{bmatrix}, K_j = \begin{bmatrix} -1 & -1 \end{bmatrix}.$$

By selecting different positive definite matrices $Q_i$, $Q_j$, the unique solution of the Lyapunov equation is given as follows

$$Q_i = \begin{bmatrix} 1 & 0 \\ 0 & 1 \end{bmatrix}, P_i = \begin{bmatrix} 0.5 & 0 \\ 0 & 0.5 \end{bmatrix}, Q_j = \begin{bmatrix} 1 & -0.5 \\ -0.5 & 1 \end{bmatrix}, P_j = \begin{bmatrix} 0.5 & 0.25 \\ 0.25 & 0.75 \end{bmatrix}$$

$L_i$ is chosen to ensure that $R_i^\mu = 0$ with $R_i = (I - L_iC_i)e^{A_i\Delta}$ ([46]). We know that $\alpha = 0.2$, $\beta = 27.7$, $\mu = 2$. $\varrho < 0.4$ must be satisfied in order for (40) to become a decreasing energy equation, picking for $\varrho = 0.2$, Lemma 4 yields $\overline{\Delta} = 0.12$. We let $\Delta = 0.1$, $\zeta = 2$ and the remaining variables can be easily calculated.

We assume that the DoS attacks applied to the measurement channel satisfy $\frac{1}{T_{\mathcal{D}}^m} = 0.15$ and $\frac{1}{\tau_{\mathcal{D}}^m} = 0.05$, the DoS attacks applied to the control channel satisfy $\frac{1}{T_{\mathcal{D}}^c} = 0.6$ and $\frac{1}{\tau_{\mathcal{D}}^c} = 0.4$, By Theorem 1, a feasible solution $\tau_\sigma = 110 \gg \tau_d = 1.14$ can be found and the switching mode is presented in Figure 3. Figures 4 and 5 depict DoS attacks on the measurement channel and the control channel, respectively. The state response of the switched system with the active control strategy under the above-mentioned situations is depicted in Figure 6.

The state response with a ZOH controller ([17,35,36]) under the same conditions is depicted in Figure 7. Comparing Figures 6 and 7, it can be observed that thanks to the predictive control sequences, the active control strategy proposed in this paper exhibits better resilience and robustness against two-channel DoS attacks and asynchronous switching behaviors.

In addition, the state response of the non-switching subsystem j with the active control strategy under the two-channel DoS attacks is depicted in Figure 8. The jamming rates

applied to the control and the measurement channels are approximately 60% and 45%, respectively, further validating Corollary 1.

In the Lyapunov method, the energy function of a system is divided into functions associated with energy reduction and energy increase. When the overall energy of the system decreases more than it increases, the system achieves stability.

In passive strategies [17,35,36], a system under DoS attacks, a system under asynchronous switching and a system under both DoS attacks and asynchronous switching are all described as functions associated with energy increase. However, in active control strategies, thanks to the predictor and the buffer, a system under DoS attacks can be described as function associated with energy reduction, while only system under asynchronous switching is described as function associated with energy increase. Therefore, in the same conditions, active control strategies exhibit greater energy reduction compared to passive control strategies. This energy reduction contributes to higher stability, resulting in superior robustness and resilience against DoS attacks and asynchronous switching behaviors.

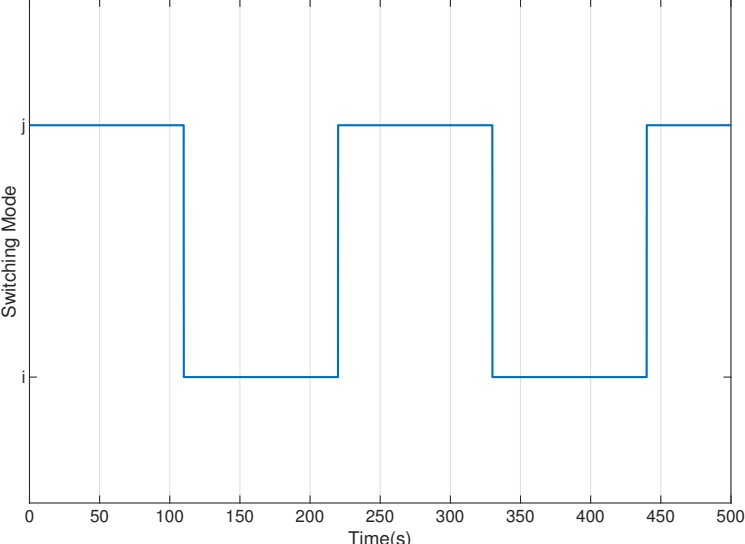

**Figure 3.** Switching mode with $\tau_\sigma = 110$.

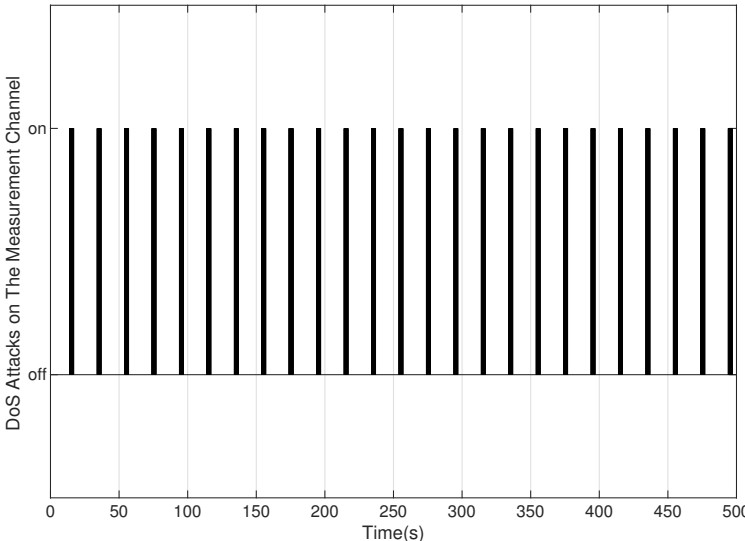

**Figure 4.** DoS attacks on the measurement channel with approximately 15% of the jamming rate.

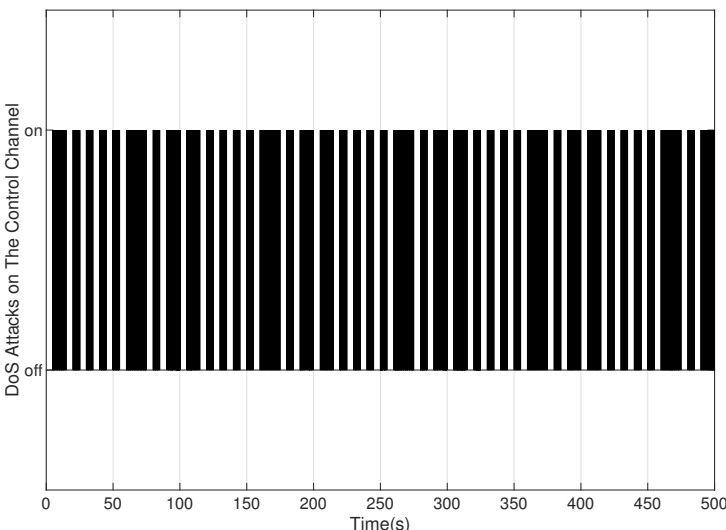

**Figure 5.** DoS attacks on the control channel with approximately 60% of the jamming rate.

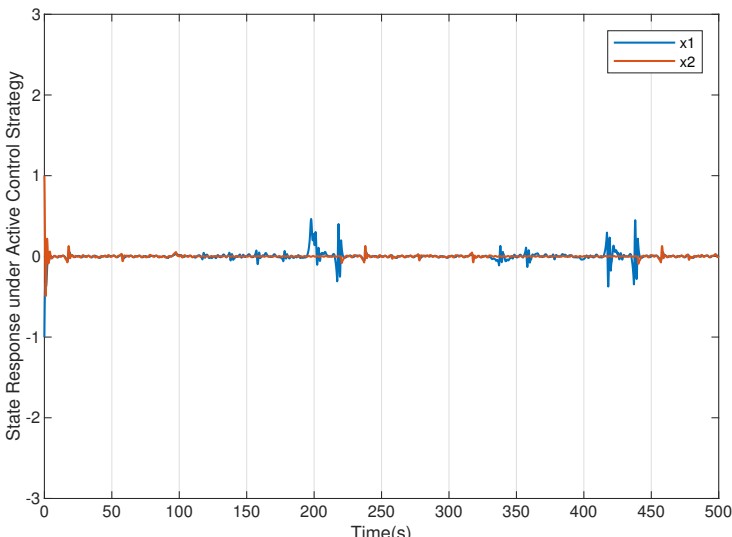

**Figure 6.** State response under the active control strategy, set the initial state $x(t_0) = [1, -1]^\mathsf{T}$ and the disturbance magnitude to 0.01.

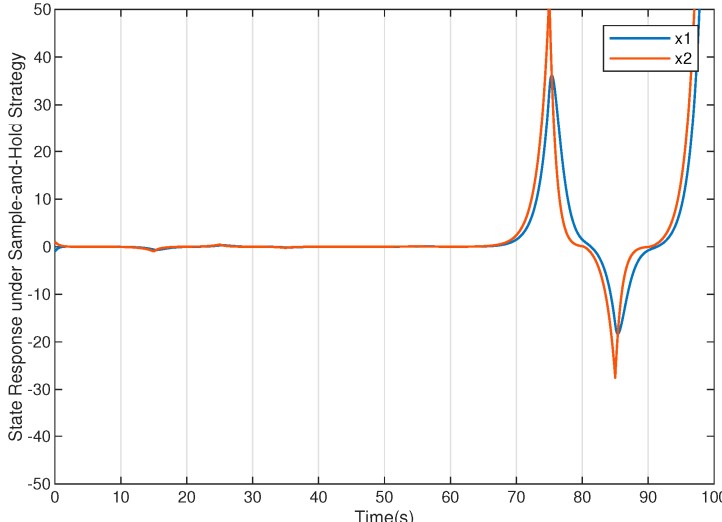

**Figure 7.** State response with a ZOH controller.

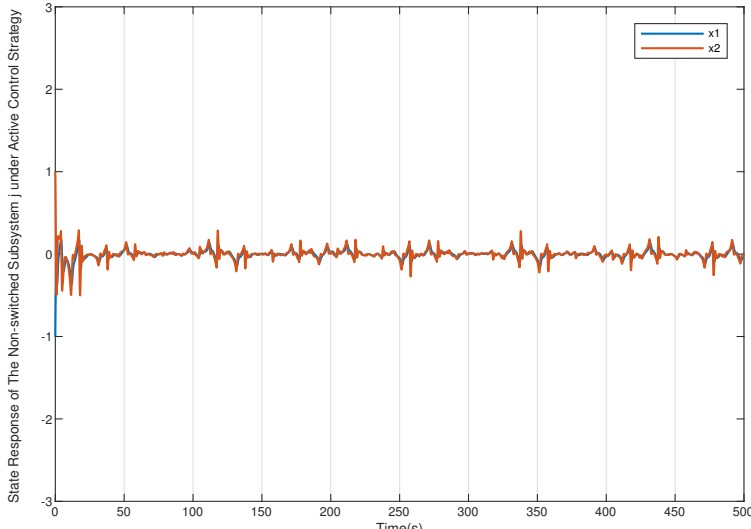

**Figure 8.** State response of the non-switched subsystem j under the active control strategy.

## 6. Conclusions

In our work, the security issue of the networked switched systems under two-channel DoS attacks has been studied. An observer-based active control strategy has been devised. With this strategy, the ISS sufficient conditions for the networked switched system under asynchronous DoS attacks have been derived. In particular, the coupling relationship among the features of two-channel DoS attacks and the switching frequency is revealed. Compared to the ZOH controller, the proposed active control strategy shows better robustness and resilience against two-channel asynchronous DoS attacks and asynchronous switching behaviors. Based on the framework in this paper, event-triggered control and nonlinear dynamics can be further investigated in the future, and the strategy can be further optimized and extended to handle more complex and sophisticated cyber threats.

**Author Contributions:** Conceptualization, J.Y.; methodology, J.Y.; software, J.Y.; validation, J.Y. and A.L.; formal analysis, J.Y.; writing—original draft preparation, J.Y.; writing—review and editing, J.Y. and A.L. All authors have read and agreed to the published version of the manuscript.

**Funding:** This research received no external funding.

**Data Availability Statement:** Data sharing is not applicable to this article.

**Conflicts of Interest:** The authors declare no conflict of interest.

## Appendix A

**Proof of Lemma 5.** During the sampling interval $[s_m, s_{m+1})$, the dynamics of $\hat{x}(t \mid z_m)$ can be descibed as

$$
\begin{aligned}
\dot{\hat{x}}(t \mid z_m) &= A_i \hat{x}(t \mid z_m) + B_i K_i \hat{x}(t_k \mid z_m) \\
&= \Phi_i \hat{x}(t \mid z_m) + B_i K_i (\hat{x}(t_k \mid z_m) - \hat{x}(t \mid z_m)) \\
&= \Phi_i \hat{x}(t \mid z_m) + B_i K_i \hat{e}(t).
\end{aligned} \tag{A1}
$$

Similar to the proof of Lemma 4, we have

$$
\| \hat{e}(t) \| \le \varrho^* \| \hat{x}(t \mid z_m) \| \tag{A2}
$$

if the sampling interval $\Delta = t_{k+1} - t_k \leq \overline{\Delta}^*$, $k \in \mathbb{N}_{\geq 0}$,

$$\overline{\Delta}^* := \begin{cases} \left(\frac{\varrho^*}{1+\varrho^*}\right)\frac{1}{\|\Phi_i\|}, & \mu_{A_i} \leq 0 \\ \frac{1}{\mu_{A_i}}\log\left[\left(\frac{\varrho^*}{1+\varrho^*}\right)\frac{1}{\|\Phi_i\|}\mu_{A_i} + 1\right], & \mu_{A_i} > 0. \end{cases} \tag{A3}$$

Substituting (A2) into the derivative of $V_i(\hat{x}) = \hat{x}^\mathsf{T} P_i \hat{x}$, it can be obtained that

$$\dot{V}(\hat{x}(t \mid z_m)) \leq (\gamma_2 \varrho^* - \gamma_1) \parallel \hat{x}(t \mid z_m) \parallel^2.$$

Selecting the sampling interval $\Delta$ satisfying (29) in Lemma 4, where $-\gamma_1 + \gamma_2\varrho < 0$, based on the positive correlation between $\Delta$ and $\varrho$, as well as the positive correlation between $\Delta$ and $\varrho^*$, we have

$$\varrho^* \leq \varrho \tag{A4}$$

further we have

$$-\gamma_1 + \gamma_2\varrho^* < 0 \tag{A5}$$

it is obtained that

$$V(\hat{x}(t \mid z_m)) \leq e^{-2\alpha_i^*(t-s_m)}V(\hat{x}(s_m \mid z_m)) \tag{A6}$$

where $\alpha_i^* = \frac{\gamma_1 - \gamma_2\varrho^*}{2\overline{\lambda}(P_i)}$, we finally have

$$\parallel \hat{x}(t \mid z_m) \parallel \leq \sqrt{\frac{\overline{\lambda}(P_i)}{\underline{\lambda}(P_i)}}e^{-\alpha_i^*(t-s_m)} \parallel \hat{x}(s_m \mid z_m) \parallel \leq \sqrt{\frac{\overline{\lambda}(P_i)}{\underline{\lambda}(P_i)}} \parallel \hat{x}(s_m \mid z_m) \parallel. \tag{A7}$$

During the interval $[z_m, s_m)$, the actuator applies control sequence $\mathcal{U}(z_{m-1})$, we have

$$\parallel \hat{x}(t_k \mid z_{m-1}) \parallel \leq \sqrt{\frac{\overline{\lambda}(P_i)}{\underline{\lambda}(P_i)}} \parallel \hat{x}(z_m \mid z_{m-1}) \parallel. \tag{A8}$$

Similarly, within interval $[s_m, s_{m+1})$, the actuator applies control sequence $\mathcal{U}(z_m)$, we have

$$\parallel \hat{x}(t_k \mid z_m) \parallel \leq \sqrt{\frac{\overline{\lambda}(P_i)}{\underline{\lambda}(P_i)}} \parallel \hat{x}(s_m \mid z_m) \parallel \tag{A9}$$

Next, we find the relationship between $\parallel \hat{x}(s_m \mid z_m) \parallel$ and $\parallel \hat{x}(z_m \mid z_m) \parallel$. During the interval $[z_m, s_m)$, the dynamics of $\parallel \hat{x}(t \mid z_m) \parallel$ can be described as

$$\begin{aligned} \dot{\hat{x}}(t \mid z_m) &= A_i\hat{x}(t \mid z_m) + B_iK_i\hat{x}(t_k \mid z_{m-1}) \\ &= \Phi_i\hat{x}(t \mid z_m) + B_iK_i\hat{\theta}(t_k) + B_iK_i\hat{e}(t) \end{aligned} \tag{A10}$$

where $\hat{\theta}(t_k) = \hat{x}(t_k \mid z_{m-1}) - \hat{x}(t_k \mid z_m)$, $\hat{e}(t) = \hat{x}(t_k \mid z_m) - \hat{x}(t \mid z_m)$. Since we assume that the asynchronous switching intervals do not overlap, i.e., no switching occurs in $[z_{m-1}, z_m]$, then we have (28) holds true for all $t_k \in [z_m, s_m)$ even if a switching behavior occurs after $z_m$ ($\mathcal{U}(z_{m-1})$ and $\mathcal{U}(z_m)$ are independent of the switching signal after $z_m$). Similar to the Proof of Lemma 4, we can derive that for the sampling interval $\Delta$ satisfying (29) in Lemma 4, where $-\gamma_1 + \gamma_2\varrho < 0$, we have

$$\| \hat{e}(t) \| \leq \varrho^{**} \| \hat{x}(t \mid z_m) \| + \varrho^{**} 4\rho \| w_t \|_\infty \tag{A11}$$

where $\varrho^{**} \leq \varrho$.

By Lyapunov method and young's inequality, we further have

$$
\begin{aligned}
\dot{V}(\hat{x}(t \mid z_m)) &= \hat{x}(t \mid z_m)^\mathsf{T}(\Phi_i^\mathsf{T} P_i + P_i \Phi_i)\hat{x}(t \mid z_m) + 2e(t)^\mathsf{T} K_i B_i^\mathsf{T} P\hat{x}(t \mid z_m) \\
&\quad + 2\theta(t_k)^\mathsf{T} K_i B_i^\mathsf{T} P\hat{x}(t \mid z_m) \\
&\leq (-\gamma_1 + \gamma_2 \varrho^{**}) \| \hat{x}(t \mid z_m) \|^2 + \gamma_2 \rho(2 + 4\varrho^{**}) \| \hat{x}(t \mid z_m) \| \| w_t \| \\
&\leq -\frac{\gamma_1 - \gamma_2 \varrho^{**}}{2\overline{\lambda}(P_i)} V_i(\hat{x}(t \mid z_m)) + \frac{2[\gamma_2 \rho(1 + 2\varrho^{**})]^2}{\gamma_1 - \gamma_2 \varrho^{**}} \| w_t \|_\infty^2
\end{aligned}
\tag{A12}
$$

then,

$$V(\hat{x}(t \mid z_m)) \leq e^{-\theta_1(t - z_m)} V(\hat{x}(z_m \mid z_m)) + \frac{\theta_2}{\theta_1} \| w_t \|_\infty^2 \tag{A13}$$

where $\theta_1 = \frac{\gamma_1 - \gamma_2 \varrho^{**}}{2\overline{\lambda}(P_i)}$, $\theta_2 = \frac{2[\gamma_2 \rho(1 + 2\varrho^{**})]^2}{\gamma_1 - \gamma_2 \varrho^{**}}$, thus,

$$
\begin{aligned}
\| \hat{x}(t \mid z_m)) \| &\leq \sqrt{\frac{\overline{\lambda}(P_i)}{\underline{\lambda}(P_i)}} e^{-\frac{\theta_1}{2}(t - z_m)} \| \hat{x}(z_m \mid z_m) \| + \sqrt{\frac{\theta_2}{\theta_1 \underline{\lambda}(P_i)}} \| w_t \|_\infty^2 \\
&\leq \sqrt{\frac{\overline{\lambda}(P_i)}{\underline{\lambda}(P_i)}} \| \hat{x}(z_m \mid z_m) \| + \sqrt{\frac{\theta_2}{\theta_1 \underline{\lambda}(P_i)}} \| w_t \|_\infty
\end{aligned}
\tag{A14}
$$

for all $t \in [z_m, s_{m+1})$. Combining (A9) and (A14), we have

$$\| \hat{x}(t_k \mid z_m)) \| \leq \frac{\overline{\lambda}(P_i)}{\underline{\lambda}(P_i)} \| \hat{x}(z_m \mid z_m) \| + \sqrt{\frac{\theta_2 \overline{\lambda}(P_i)}{\theta_1 \underline{\lambda}^2(P_i)}} \| w_t \|_\infty \tag{A15}$$

for all $t \in [z_m, s_{m+1}]$. By continuity of $x(t)$ we have

$$
\begin{cases}
\| \hat{x}(z_m \mid z_{m-1})) \| \leq \| x(z_m) \| + \rho \| w_t \|_\infty \\
\| \hat{x}(z_m \mid z_m)) \| \leq \| x(z_m) \| + \rho \| w_t \|_\infty .
\end{cases}
\tag{A16}
$$

Combining (A8), (A14) and (A16), we finally have

$$\| \hat{x}(t_k) \| \leq \frac{\overline{\lambda}(P_i)}{\underline{\lambda}(P_i)} \| x(z_m) \| + \left( \frac{\overline{\lambda}(P_i)}{\underline{\lambda}(P_i)} \rho + \sqrt{\frac{\theta_2 \overline{\lambda}(P_i)}{\theta_1 \underline{\lambda}^2(P_i)}} \right) \| w_t \|_\infty . \tag{A17}$$

Then the proof is completed. □

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
