# Peer review of "Observer-Based Active Control Strategy for Networked Switched Systems against Two-Channel Asynchronous DoS Attacks"

_actuators, doi:10.3390/act12080335_

Round 1

Reviewer 1 Report

The authors propose an observer-based active control strategy for networked switched systems against two-channel asynchronous DoS attacks. Sufficient conditions are derived to guarantee the input-to-state stability and simulation results are offered. The article is well organized and the proposed results seem correct. However the following issues should be considered in the revised version:

1/ The motivations and innovations of this paper need to be more specifically stated in the end of the introduction. What are the innovations and differences of the active control strategy proposed in this article compared to the existing methods.

2/The algorithm and description of the active control strategy should be more detailed.

3/ The details of Corollary 1 need to be further elaborated, such as what is the design methodology for buffer size in the switching-free case.

4/ There are several language problems to be polished. Please solve them.

5/ It is better to label the transmission of the signals in Figure 1 and which signals are specifically blocked by the DoS attacks.

6/ The article also has several typos and language issues, which need to be checked and corrected in the revision.

7/The definitions of several symbols are missing.

Reviewer 2 Report

This paper addresses the security issue of the networked switched systems under two-channel asynchronous denial-of-service (DoS) attacks. My comments for the paper are:

In the abstract, you mentioned that the proposed active control strategy mitigates the negative impact of two-channel asynchronous DoS attacks on control performance and stability. Could you elaborate on how the observer-based strategy effectively handles the attacks on the measurement and control channels separately? How does it ensure that control signals can be updated even when the control channel is blocked?

Switching behavior in networked switched systems can be complex. Could you explain how the designed switching signal ensures input-to-state stability (ISS) of the system under both two-channel asynchronous DoS attacks and asynchronous switching behaviors? How do these two factors interact, and how does your approach address the challenges introduced by their coupling relationship?

Observer-based control strategies often rely on system state estimation. How accurate and reliable is the mode-dependent finite-time observer in estimating the system state, especially under the influence of DoS attacks on the measurement channel? How do you validate the effectiveness of this observer in practical scenarios?

The abstract mentions the use of a predictor and buffer in the control strategy to manage the control signals transmitted to the actuator. Could you provide more details on how the predictor and buffer are designed and how they contribute to maintaining control performance and stability during DoS attacks?

The paper discusses the active control strategy's effectiveness and merits through an example and a comparative experiment. Could you share some insights into the experimental setup and the specific metrics used to evaluate the performance of the proposed strategy in comparison to other approaches? What were the key findings from the experiments, and how do they support the claims made in the paper?

Networked systems often operate in dynamic environments with changing conditions. How does your proposed strategy handle adaptability to different types of DoS attacks, changing attack patterns, and varying network conditions? How do you ensure robustness in real-world scenarios where attack characteristics may evolve over time?

In practical applications, the resources available for implementing security measures might be limited. What are the computational and resource requirements of your proposed active control strategy, and how does it address the challenge of resource-constrained environments, especially in IoT or embedded systems?

The abstract mentions the impact of two-channel asynchronous DoS attacks on control performance and stability. Could you elaborate on any real-world implications of such attacks and how your research contributes to enhancing the security and resilience of networked systems in critical applications, such as industrial control or autonomous vehicles?

In the paper, you addressed the issue of DoS attacks, but cyber-physical systems may face a broader range of security threats. How do you envision extending or integrating your approach to address other types of cyber threats that can impact networked switched systems' safety and reliability?

Author can read the following papers to increase the technical strength of the paper: Boosting-based DDoS detection in internet of things systems, Detection of Distributed Denial of Service (DDoS) Attacks Using Computational Intelligence and Majority Vote-Based Ensemble Approach

Finally, what are some potential future research directions and advancements in this area? How can the proposed strategy be further optimized or extended to handle more complex and sophisticated cyber threats, and what are the key challenges to be addressed in this direction?

Moderate editing of English language required

Round 2

Reviewer 2 Report

The author address all the previous comments but still some issues are pending.

Check my previous comments

The abstract needs to be rewritten to point out significance and impact of the paper.

In the related work, it is recommended to refer the contribution made by the researchers and the novelty of the research. However, the author does not mention that.

I recommend that the authors add some more current articles to improve the paper's overall quality. The preparation of a comparative analysis of the current publications on this subject should also be included.

Avoid presenting with lengthy paragraph.

Paper needs to polish and provide a detailed explication of theoretical/systematic aspects behind this paper.

Minor editing of English language required
